# HHW-Ego: DSLR-Quality Enhancement for Multi-Source Wearable Ego Imaging

## Abstract

Wearable ego-centric images are now in high demand for scenarios ranging from daily smart glass usage to embodied intelligence. But the image quality is far behind smart phone due to the lacking paired high-quality reference images and the dedicated enhancement systems. To solve this, a customized degradation pipeline is designed to generate paired samples matching wearable camera images, boosting the upper limit of enhancement performance. Besides a two-stage enhancement framework is further built: first, tuning an efficient model on the paired dataset to enhance real wearable images; second, using hyperspectral data to refine color temperature for better quality. Moreover we present HHW-Ego, a multi-source paired dataset integrating with Hyperspectral, High-dynamic range and Wearable Ego-centric data, which includes hundreds of image groups spanning indoor/outdoor and day/night scenes. Experiments show the framework effectively enhances images of varying quality, and matches DSLR camera quality in specific scenarios.

## 1 Introduction

Wearable ego-centric imaging has recently drawn increasing attention due to its wide range of applications, including daily smart-glass usage, life-logging, and embodied intelligence for robotics and AR/VR systems (Betancourt et al., 2015; Ragusa et al., 2020; Grauman et al., 2022; Yang et al., 2025b). Compared with traditional third-person cameras, wearable devices offer a first-person perspective that captures unique information about user activities and environments. Such capabilities make them valuable for human–computer interaction, navigation, and downstream embodied intelligence tasks (Zhang et al., 2018b; Broxton et al., 2020). Despite this potential, the visual quality of wearable ego-centric images remains significantly behind that of smartphone or DSLR cameras, which greatly limits their utility in both consumer and research settings. The primary challenge stems from the intrinsic hardware limitations of wearable cameras. Compact form-factors, limited optics, low dynamic range, and inaccurate color rendition penalize wearable systems (Chen et al., 2018). Furthermore, unlike smartphones, which benefit from large-scale paired datasets and dedicated imaging pipelines, ego-centric cameras lack high-quality reference images for supervised training. This absence of paired supervision hinders the development of effective enhancement models, leaving most wearable imagery unsuitable for high-level vision tasks or immersive user experiences.

In recent years, there have been significant advances in super-resolution (SR) and image enhancement that are relevant to this problem. For example, SwinIBSR (Shi et al., 2024) proposes a Transformer-based method for real-world infrared image super-resolution, with a novel degradation model and mixed training to better generalize across visible and infrared domains. (Jin et al., 2024) also shows how physics-enhanced networks can reconstruct high-resolution from compressive measurements over large scenes. Moreover, in image enhancement under novel optics, researchers have used deep learning to boost image quality of metalens cameras by learning paired low- vs high-quality images to recover fine detail and reduce optical blur (Seo et al., 2024). These methods show that with the right degradation modeling and paired supervision, SR/enhancement performance can be dramatically improved. HVI (Yan et al., 2025) introduces a new color space for low-light improvement, GaussHDR (Liu et al., 2025a) combines 3D Gaussian splatting with HDR reconstruction, and UHD-processer (Liu et al., 2025b) unifies UHD image restoration. Besides (Feijoo et al., 2025) focus on the restoration of low-light infrared images.

In the hyperspectral domain, there have also been promising developments. Deep learning-based hyperspectral image reconstruction (from RGB to HSI) has been used for agri-food quality assessment, demonstrating that reconstructed spectra can approximate ground truth well, enabling downstream tasks (Ahmed et al., 2024). There is also a comprehensive review surveying hyperspectral image reconstruction methods from compressive measurements, which underscores the importance of recovering both spectral and spatial fidelity (Han et al., 2025). Additionally, algorithms for fusing high spatial resolution multispectral images with low spatial resolution hyperspectral images (spatial-spectral feature fusion) have been developed to yield high-spatial resolution hyperspectral output (Yi et al., 2024). Such methods are relevant because improving color fidelity, spectral consistency, and detail are all critical for wearable ego-centric imaging, especially under challenging illumination.

To address these issues specifically for wearable ego-centric images, we propose a two-stage enhancement framework. First, we design a customized degradation pipeline that synthesizes realistic wearable-like data from high-quality sources, thereby generating paired samples that bridge the gap between reference-rich domains and wearable devices. This enables efficient training of enhancement models on data that faithfully mimics real-world degradations. Second, we leverage hyperspectral data to refine color temperature and achieve perceptually consistent color reproduction, which is particularly critical for scenes with complex illumination. Together, these strategies boost the upper limit of enhancement quality and bring wearable imagery closer to smartphone- and DSLR-level fidelity. Beyond the framework itself, we introduce HHW-Ego, a novel multi-source paired dataset that integrates Hyperspectral, High-dynamic range (HDR), and Wearable Ego-centric data. The dataset contains hundreds of image groups spanning diverse indoor/outdoor and day/night scenes, offering comprehensive coverage for both supervised enhancement and generalization analysis. Experiments show that our method not only enhances wearable images of varying qualities but also approaches DSLR camera quality in certain scenarios. We believe that HHW-Ego, together with our proposed pipeline, will serve as a strong foundation for future research on wearable image enhancement, perception, and embodied intelligence.

Our contributions can be summarized as follows:

- We identify the fundamental limitations of wearable ego-centric imaging and design a customized degradation pipeline to generate realistic paired training samples.

- We develop a two-stage enhancement framework that couples efficient supervised training with hyperspectral-guided color refinement.

- We construct HHW-Ego, the first multi-source paired dataset integrating hyperspectral, HDR, and wearable ego-centric imagery across diverse environments.

- We demonstrate through extensive experiments that our approach significantly improves wearable image quality and narrows the gap with DSLR cameras.

## 2 RELATED WORK

CNN-based and GAN-based super-resolution works—represented by BSRGAN (Zhang et al., 2021) and Real-ESRGAN (Wang et al., 2021)—typically adopt a two-component architecture: CNNs serve as the backbone to extract and reconstruct image features, while GANs (as proposed in (Goodfellow et al., 2020)) are integrated mainly as a post-processing module or loss function to enhance visual quality. The CNN backbone, with its efficient hierarchical feature extraction capability, lays the foundation for recovering fine details from low-quality images; GANs, meanwhile, optimize the output by minimizing the distribution gap between generated and real high-quality images—often via adversarial loss—to refine texture and contrast. This design offers notable advantages for industrial scenarios: the CNN-based framework ensures computational efficiency (reducing inference latency), and the GAN-driven optimization avoids over-smoothing common in traditional SR methods. These traits make such models widely applicable in streaming media, real-time video enhancement, and other latency-sensitive tasks where rapid processing is prioritized. However, the introduction of GANs also brings inherent limitations: the adversarial training process may generate unrealistic textures or artifacts (e.g., unnatural edge halos, inconsistent color patches), which, while improving objective metrics like PSNR in some cases, restrict subjective quality—especially for wearable ego-centric images where structural consistency and perceptual naturalness (as evaluated

by LPIPS (Zhang et al., 2018a)) are critical for downstream tasks like embodied interaction or AR overlay.

However, the practical large-scale application of Transformer- and Diffusion-based super-resolution methods in industry is hindered by prohibitive computational costs, which stem from two core limitations. Firstly, Transformer-induced computational overhead as PASD (Yang et al., 2024) and PFT-SR (Long et al., 2025): the self-attention mechanism in Transformer-based modules—adopted in methods like PASD exhibits quadratic time complexity (O ($n^2$), where n denotes the number of image tokens). This characteristic leads to enormous memory consumption and computational demands when processing high-resolution wearable ego-centric images (e.g., 1080p and above), rendering real-time inference infeasible on resource-constrained wearable devices such as smart glasses equipped with low-power GPUs. Secondly, Diffusion sampling latency like StableSR (Wang et al., 2024) and SUPIR (Yu et al., 2024): Diffusion-based components, as seen in StableSR and PFT-SR, typically require dozens of iterative sampling steps to generate high-quality enhanced images. Even with step-reduction optimizations, the resulting inference latency remains substantially higher than that of CNN-GAN methods, failing to meet the real-time performance requirements of critical industrial scenarios—including live streaming, real-time AR overlay for smart glasses, and low-latency feedback loops in wearable device-driven embodied intelligence tasks.

## 3 METHODOLOGY

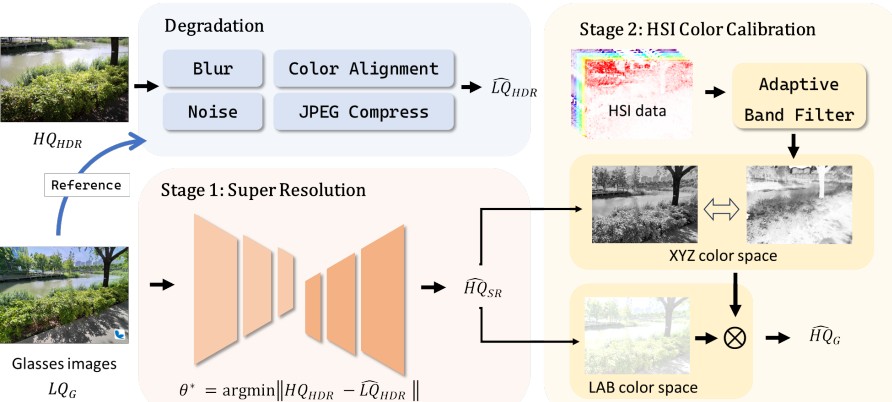

Figure 1: Overview of our pipeline. Black solid lines represent the degradation process, black dashed lines represent the model training process, and red solid lines represent the inference process.

### 3.1 OVERVIEW

Our data pipeline consists of acquisition, preprocessing, degradation, and enhancement as shown in **Figure 2 (b)** and **Figure 1**. We captured paired wearable-device images, HDR images, and hyperspectral (HSI) images across over 300 indoor and outdoor scenes under both daytime and nighttime conditions. Due to differences in camera parameters and viewing angles, we first perform cross-modal image registration to ensure approximate semantic alignment across the three sets.

To enhance wearable-device images, we train a super-resolution reconstruction model capable of enhancing details in low-resolution wearable-style images. To obtain paired low-quality (LQ) and high-quality (HQ) images for training, we degrade HDR images while aligning their style and noise characteristics with wearable-device images in both Lab and RGB color spaces. The resulting paired data is used to train an SR model $\theta$, forming the first stage of enhancement, which not only restores fine details but also adapts the wearable images towards HDR-style quality. In the second stage, we leverage HSI information to further enhance color and detail in the visible spectrum.

## 3.2 ALIGNED DEGRADATION

Conventional super-resolution models aim to restore high-frequency details that are lost during image downsampling. In practice, the degradation process is typically simulated by applying blur and noise to high-quality images, where randomized blur kernels and noise levels are often introduced to enhance model robustness as (Damera-Venkata et al., 2000; Pei et al., 2019). However, such synthetic degradations rarely reflect the true distribution of real-world wearable images, leading to a mismatch between training and deployment conditions.

Formally, let the real-world LQ image be denoted as $LQ_{\text{source}}$ and the corresponding HQ target as $HQ_{\text{source}}$. We aim to learn a mapping $\theta$ such that

$$\theta(LQ_{\text{source}}) \approx HQ_{\text{source}}. \tag{1}$$

In standard blind SR settings, a degraded image is generated as

$$LQ_{\text{target}} = (HQ_{\text{target}} \otimes k) \downarrow_s + n, \tag{2}$$

where $k$ denotes the blur kernel, $\otimes$ represents convolution, $\downarrow_s$ indicates downsampling by factor $s$, and $n$ is additive noise. This synthetic degradation does not guarantee alignment between $LQ_{\text{target}}$ and the real $LQ_{\text{source}}$.

To better adapt $\theta$ to wearable imagery, we define an *aligned degradation process*:

$$LQ_{\text{target}} = (\text{Lab}_s(HQ_{\text{target}}) \otimes \hat{k}) \downarrow_s + \hat{n}, \tag{3}$$

where $\text{Lab}_s(\cdot)$ performs Lab-space style alignment, $\hat{k}$ is a blur kernel estimated to match $LQ_{\text{target}}$ with $LQ_{\text{source}}$, and $\hat{n}$ is noise modeled from the source domain.

Overall, the Blind SR task can be formulated as a maximum a posteriori (MAP) estimation:

$$\arg\max_x p(y \mid x, s, k, n)\, p(x)\, p(s, k, n), \tag{4}$$

where $p(x)$ denotes the prior over HR images, and $p(k)$ serves as the alignment prior. The aligned degradation model assumes independence among $s$, $k$, and $n$.

## 3.3 TWO-STAGE MULTIMODAL ENHANCEMENT

Enhancement of wearable-device images combines detail reconstruction and color correction in a unified framework. In the first stage, we fine-tune a super-resolution model on paired $HQ_{\text{target}}$ and $LQ_{\text{target}}$ images obtained from the aligned degradation, enabling the model to restore fine structures while adapting to the style of the wearable devices. In the second stage, we leverage hyperspectral (HSI) data to further refine colors in the visible range by selecting informative spectral bands and applying correlated color temperature (CCT) mapping, producing enhanced images that are both detailed and color-consistent with the source scenes.

### 3.3.1 HYPERSPECTRAL BAND SELECTION

Hyperspectral and RGB images often suffer from illumination color casts, spectral distortions, and overexposed regions. To enhance color fidelity, we perform a three-stage correction pipeline: white balance correction, HSI color correction, and overexposure recovery. Given an image cube $I \in \mathbb{R}^{H \times W \times C}$, each pixel spectrum $s_i \in \mathbb{R}^C$ is processed individually or in local neighborhoods.

**White Balance Correction.** Given an HSI or RGB image $I$, we first estimate the scene illuminant $\mathbf{L} = [L_R, L_G, L_B]$ by averaging high-intensity pixels or using a reference white region. The white-balanced image is computed as

$$I'_c = \frac{I_c}{L_c} \cdot \bar{L}, \quad c \in R, G, B, \tag{5}$$

where $\bar{L}$ is the channel-wise normalization of $L_c$. This reduces color casts caused by illumination differences.

**HSI Color Correction.** For each pixel spectrum $s_i \in \mathbb{R}^C$, a target color reference $\hat{s}i$ is defined based on statistical matching or histogram alignment. The corrected spectrum is obtained by

$$s'i = (s_i - \mu_s) \cdot \frac{\sigma\hat{s}}{\sigma_s} + \mu\hat{s}, \tag{6}$$

where $(\mu_s, \sigma_s)$ and $(\mu_{\hat{s}}, \sigma_{\hat{s}})$ denote the mean and standard deviation of the source and reference bands, respectively. This operation compensates for color drifts across the spectral channels.

**Overexposure Recovery.** For each pixel, an overexposure mask $M$ is defined as

$$M = \max(I_X, I_Y, I_Z) > 1.0, \tag{7}$$

where $I_X, I_Y, I_Z$ are the XYZ channels. The mask $M$ is then dilated to form an outer ring $R$, and only sufficiently bright pixels are retained:

$$R_{\text{corrected}} = \text{mean}(I * R). \tag{8}$$

Finally, pixel values in $R_{\text{corrected}}$ are adjusted using local neighborhood statistics or interpolation from valid neighboring pixels to restore saturated colors.

### 3.3.2 HSI TO CCT-GUIDED RGB CORRECTION

The proposed algorithm performs an efficient CCT-based color correction by jointly leveraging hyperspectral and RGB information. First, a subset of hyperspectral bands is converted into the CIE XYZ color space, from which a pixel-wise CCT map is estimated. This CCT map serves as the target illumination reference, effectively capturing the fine-grained spectral characteristics of the hyperspectral data. In parallel, the input RGB image is converted to XYZ and its original CCT is computed. By taking the ratio between the target and original CCTs, a correction factor is derived and adaptively applied to the XYZ representation of the RGB image. This adjustment efficiently transfers illumination consistency from the hyperspectral domain to the RGB domain, achieving robust color fidelity. Finally, the corrected XYZ is transformed back into RGB space. Through this procedure, the algorithm provides an effective fusion strategy that exploits hyperspectral priors to enhance RGB imagery with minimal computational overhead. And the corresponding algorithm is as following to get $I_{\text{rgb\_corrected}}$:

---

**Algorithm 1** CCT-based Color Correction from HSI to RGB

---

**Require:** HSI bands $H_{\text{sub}}$, RGB image $I_{\text{rgb}}$, correction strength strength
**Ensure:** Corrected RGB image $I_{\text{rgb\_corrected}}$
 1: Convert $H_{\text{sub}}$ to CIE XYZ using standard color matching functions in (CCUP, 1931)
 2: Compute pixel-wise CCT map of $T_{\text{target}}$
 3: Convert $I_{\text{rgb}}$ to XYZ and compute original CCT of $T_{\text{original}}$
 4: Compute CCT adjustment ratio for each pixel $p$:

$$R_{\text{cct}}(p) = \left( \frac{T_{\text{target}}(p)}{T_{\text{original}}(p)} \right)^{\text{strength}}$$

 5: Adjust XYZ channels by $R_{\text{cct}}$:

$$XYZ_{\text{adj}} = XYZ_{\text{rgb}} \cdot R_{\text{cct}}$$

 6: Convert $XYZ_{\text{adj}}$ back to RGB:

$$I_{\text{rgb\_corrected}} = \text{XYZ\_to\_RGB}(XYZ_{\text{adj}})$$

 7: Clip $I_{\text{rgb\_corrected}}$ into $[0, 1]$ and save result

---

## 4 DATASET

### 4.1 MULTI-SOURCE DATA COLLECTION

To address the inherent limitations of imaging quality in Wearable selfie devices (such as smart glasses), we constructed the HHW-Ego (Hyperspectral-HDR-Wearable Ego-centric dataset), which is a multimodal and multi-source paired dataset for wearable visual enhancement tasks with varying data description as shown in Table 1. HHW-ego integrates HSI, HDR, and wearable first-person perspective (ego-centric) images, aiming to provide high-quality reference signals and support end-to-end enhancement training from low-quality wearable images to high-quality perception-consistent images .

Figure 2: (a) Schematic diagram of the experimental setup for the HHW-Ego data acquisition platform. (b) Schematic diagram illustrating the pipeline for image co-registration and sub-pixel target enhancement

Table 1: Multi-source camera characteristics.

| Devices | Raw Size | Data Description |
|---|---|---|
| EmdoorVR Glasses | $4000 \times 3000 \times 3$ | RGB(8bit) |
| RayNeo V3 AI Glasses | $3864_{\pm 168} \times 2760_{\pm 264} \times 3$ | RGB(8bit) |
| Canon EOS 77D DSLR | $6000 \times 4000 \times 3$ | HDR(14bit) |
| Seetrum Hyperspectral Sensor | $1184 \times 1600 \times 31$ | 380 - 980 nm |

**Wearable Device Images:** The Qualcomm AR1 platform paired with SONY IMX681 sensor is a mainstream high-end smart glasses solution. It processes images through a complex image signal pipeline(ISP), converting Bayer's raw data into the final polished RGB image. We use current consumer-grade wearable vision systems EmdoorVR Glasses [1] and RayNeo AI Glasses [2] in **Figure 2** to capture the corresponding images.

**High Dynamic Range Image (HDR):** Use professional-grade single-lens reflex cameras (such as Canon EOS 77D) to capture multiple frames of exposure images at the same time point and from the same perspective, and generate high dynamic range images through HDR synthesis algorithms. HDR images retain rich details from dark to bright areas, serving as high-quality visible light domain references for subsequent super-resolution and detail reconstruction tasks.

**Hyperspectral Image (HSI):** The spectral cube data of the scene was collected using a snapshot spectral imager (Seetrum Hyperspectral Sensor) [3]. Each pixel contains a continuous spectral response within the 380-980 nm band range, providing rich physical color information. All acquisition processes adopt a fixed tripod + synchronous trigger mechanism to ensure the spatial consistency of the three modal images. Meanwhile, detailed metadata such as timestamps, light intensity, camera parameters (focal length, shutter speed), ambient temperature and geographical location should be recorded to facilitate subsequent analysis and modeling.

### 4.2 Image Registration

Due to the significant differences among various sensors in terms of field of view (FOV), focal length, distortion characteristics, etc., direct splicing or comparison may lead to spatial misalignment. Therefore, after collection, fine cross-modal registration needs to be performed. The center area of the LQ Image (Image 1 in **Figure 2** b) collected by EmDoorVR should be cropped and scaled to the same resolution as the HDR image (Image 2 in **Figure 2**b) to reduce the computational load and focus on the overlapping area. The SIFT (Scale-Invariant Feature Transform) algorithm is used to extract key points in the overlapping area between LQ and HDR images, and the initial correspondence relationship is established. Based on the matching point pairs, the initial transfor-

---

[1]https://vr.emdoor.com/

[2]https://www.rayneo.cn

[3]https://www.seetrum.com/

mation model (such as affine transformation or perspective projection matrix) is estimated, and the HDR image and HSI image are mapped to the LQ image coordinate system to achieve initial spatial alignment. Meanwhile, the hyperspectral data is uniformly sized to the cropped size of LQ Image, and pixel region-level color enhancement is performed. Finally, 300 sets of strictly aligned three-channel image sequences are output - LQ, HQ, and HSI. Ensure that all images are highly consistent in terms of space, semantics and pixel-level details to meet the data requirements of tasks such as multimodal fusion, super-resolution reconstruction and spectral enhancement.

## 5 EXPERIMENTS

### 5.1 METRICS

Given the absence of paired ground-truth data for real ego-centric wearable images, we adopt a no-reference (NR) image quality assessment protocol, aligning with practical deployment. Our core evaluation prioritizes three NR metrics: NIQE (Mittal et al. (2012b)) assesses the enhanced image's naturalness (preventing over-enhancement artifacts); BRISQUE (Mittal et al. (2012a)) quantifies the correction of spatial distortions like motion blur; and PIQE (Venkatanath et al. (2015)) evaluates local quality on a perception-based patch level. To confirm the structural preservation essential for downstream embodied intelligence tasks, we additionally employ the Structural Similarity Index (SSIM) (Wang et al. (2004)) as a supplementary reference, which validates the retention of inherent scene structures against the original input. This combined approach ensures a comprehensive evaluation of both real-world quality and structural fidelity.

### 5.2 IMPLEMENTATION

Our degradation pipeline simulates realistic wearable ego-centric images by fitting blur kernels with KernelGAN (Bell-Kligler et al., 2019) and adding combined Gaussian–Poisson noise, where Gaussian parameters are estimated from the wearable images and Poisson parameters are randomly sampled from [0.02, 0.1]. The RealESRGAN generator with $\mathcal{L}_1$ loss function is fine-tuned on HDR images paired with these degraded LQ images, using a learning rate of 0.1 with the Adam optimizer for 1000 iterations. All fine-tuning and inference are conducted on an NVIDIA A100 GPU, while HSI-guided color calibration and overexposure correction are applied in a training-free manner.

### 5.3 COMPARISON WITH STATE OF ART

As shown in Table 2, we evaluate no-reference image quality metrics, with higher SSIM and lower NIQE, BRISQUE, and PIQE reflecting better perceptual quality. Our methods exhibit notable advantages in perceptual performance: Specifically, Ours (PFT based) achieves the lowest NIQE (5.5244) and BRISQUE (19.3457) across all methods, and its PIQE (45.9188) is highly competitive with state-of-the-art approaches like PFT-SR. Compared to baseline methods (e.g., BSRGAN, Real-ESRGAN), our variants consistently reduce perceptual metric values—for example, Ours (PFT based) decreases NIQE by over 1 point relative to BSRGAN and nearly 1 point relative to StableSR, while lowering BRISQUE by more than 5 points compared to BSRGAN. Even against advanced methods (PASD, SUPIR, PFT-SR), our schemes (especially those based on PFT and SUPIR) either outperform or match them in perceptual metrics. This demonstrates that our approach effectively boosts perceptual image quality, which is more aligned with human visual perception.

As qualitatively shown in Figure 3, a visual comparison of methods reveals distinct performance trade-offs. GAN-based blind super-resolution methods (e.g., BSRGAN, RealESRGAN) sacrifice detail by producing results that are overly smooth. Conversely, Diffusion-based solutions (e.g., StableSR, SUPIR) achieve greater sharpness but frequently introduce structural distortions or visible artifacts that deviate from the original scene. Our method is built upon a foundational SR backbone and integrates HSI data for targeted color correction and over-exposure recovery, ensuring structural consistency while simultaneously achieving enhanced color realism.

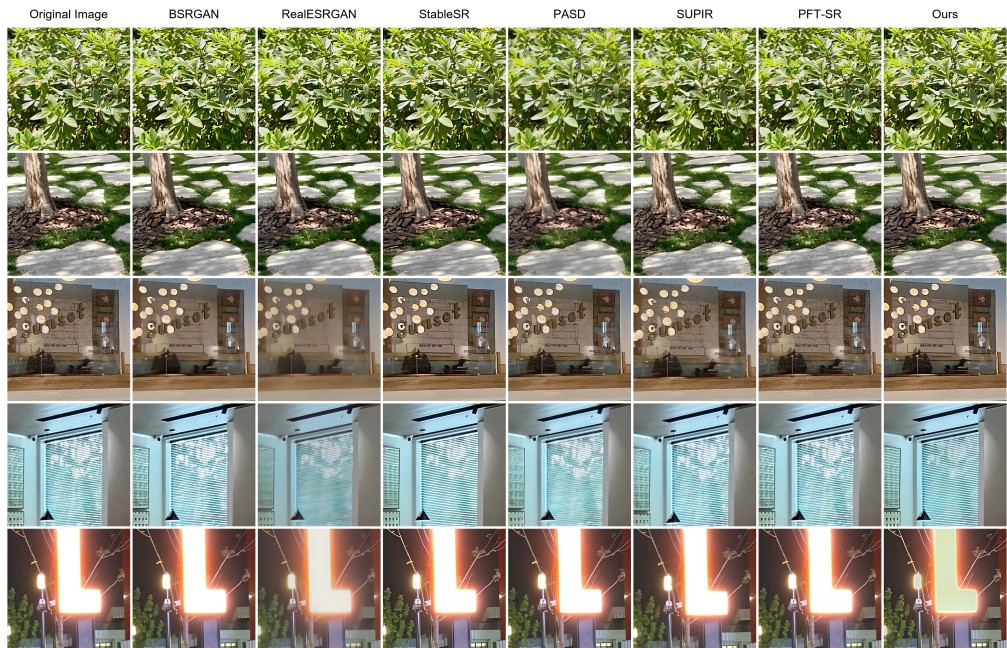

Figure 3: Visual comparison with state-of-the-art super-resolution methods on diverse real-world scenes.

Table 2: Comparison of no-reference image quality metrics. Lower NIQE, BRISQUE, and PIQE indicate better perceptual quality.

| Method | SSIM ↑ | NIQE ↓ | BRISQUE ↓ | PIQE ↓ |
|---|---|---|---|---|
| BSRGAN (Zhang et al., 2021) | 0.4299 | 6.2345 | 24.3822 | 55.0286 |
| Real-ESRGAN (Wang et al., 2021) | 0.4456 | 6.4384 | 24.2635 | 57.5201 |
| StableSR (Wang et al., 2024) | 0.4152 | 6.1689 | 21.4317 | 50.8174 |
| PASD (Yang et al., 2024) | 0.4390 | 6.0219 | 24.0410 | 52.6878 |
| SUPIR (Yu et al., 2024) | 0.4149 | 5.7492 | 21.2885 | 48.6881 |
| PFT-SR (Long et al., 2025) | 0.4284 | 5.7156 | 20.1179 | 45.4049 |
| Ours (Real-ESRGAN based) | 0.4165 | 6.0919 | 23.5923 | 55.2639 |
| Ours(PASD based) | 0.4291 | 5.9181 | 23.4390 | 56.0496 |
| Ours (SUPIR based) | 0.4090 | 5.6741 | 20.7979 | 51.5428 |
| Ours (PFT based) | 0.4131 | 5.5244 | 19.3457 | 45.9188 |

## 5.4 USER STUDY

Since no-reference metrics and SSIM still fail to represent true subjective quality, we recruited 5 participants in the laboratory to conduct human visual subjective evaluation, performing pairwise comparisons for BSRGAN, Real-ESRGAN, StableSR, PASD, SUPIR, PFT-SR, and our method respectively. Our scheme has an absolute advantage over BSRGAN and Real-ESRGAN (with an approximate 95% probability that our method is preferred), and a distinct advantage when compared with StableSR, PASD, SUPIR, and PFT-SR (with an approximate 75% probability that our scheme is preferred).

## 5.5 ABLATION STUDY

**Ablation Study on Modules:** To evaluate the contributions of each component in our pipeline, we conduct ablation experiments on three configurations: (1) baseline SR, (2) SR enhanced with hyperspectral information, and (3) SR with both hyperspectral guidance and exposure correction. Table 3

reports the performance using several no-reference image quality metrics. The results demonstrate a clear incremental improvement. Incorporating hyperspectral information into the SR model consistently enhances detail preservation and color fidelity compared to the baseline SR. Further adding exposure correction provides additional gains, particularly in visual consistency and dynamic range, leading to the best overall performance. These findings validate that both hyperspectral guidance and exposure correction contribute positively and stably to the restoration quality.

Table 3: Objective quality metrics for wearable-device images under different enhancement methods. Lower values indicate better perceptual quality.

| Method | NIQE ↓ | BRISQUE ↓ | PIQE ↓ |
|---|---|---|---|
| SR enhancement | 6.1946 | 24.0594 | 53.9674 |
| SR + Hyperspectral enhancement | 6.1307 | 24.0953 | 55.6191 |
| SR + Hyperspectral + Exposure correction | 6.0919 | 23.5923 | 55.2639 |

**Ablation Study on Degradation Configurations:** We conducted ablation experiments to investigate the impact of We performed ablation studies to determine which combinations of color, blur, and noise best mimic real wearable imagery. As summarized in Table 3, varying blur kernels and noise levels significantly impacts image quality metrics. Based on these observations, we found that tailored degradation strategies are crucial for matching real-world conditions. For example, the align-kernel-random configuration yielded the best visual fidelity for low-light nighttime scenes, while align-random-random provided more consistent results for overexposed conditions. These findings underscore the necessity of scene-specific degradation parameters.

Table 4: Ablation study on degradation alignment. LPIPS and FID values are lower for better quality, while SSIM and PSNR are higher for better quality.

| Color | Blur | Noise | LPIPS ↓ | FID ↓ | SSIM ↑ | PSNR ↑ |
|---|---|---|---|---|---|---|
| x | random | random | 0.2239 | 51.8279 | 0.4269 | 13.9433 |
| align | random | random | 0.2201 | 51.1615 | 0.4244 | 14.0457 |
| align | kernel | random | 0.2202 | 50.9877 | 0.4262 | 14.0682 |
| align | kernel | align | 0.2202 | 50.9031 | 0.4258 | 14.0620 |

**Ablation Study on Generalization:** To further compare the generalization ability of our method, we conducted comparisons on data from multiple sources, including the widely-used RayNeo glasses. As depicted in Figure 5, our method consistently improves image quality across diverse scenes (Outdoor, Indoor, Night) and data sources (EmdoorVR, RayNeo). Specifically, our approach effectively reduces noise in challenging indoor and nighttime scenarios, leading to sharper textures. Furthermore, by integrating HSI data with super-resolution, we successfully mitigate overexposure in bright areas, recovering highlight information and significantly improving color accuracy. These results confirm the robust generalization ability of our method, demonstrating consistent quality enhancement across various environments and wearable devices.

## 5.6 GAMUT ANALYSIS

We performed gamut analysis in both the xy chromaticity space and the Lab color space for the images before and after the HSI-based correction, in order to further demonstrate the advantage of our method in expanding the effective color gamut. The $xy$ space is composed of the normalized chromaticity coordinates $(x, y)$, which represent the distribution of chromatic content independent of luminance, and therefore reveal the global color rendition and chromatic shift characteristics of the image. The $ab$ plane of the Lab space is composed of opponent-color channels $a$ (green–red) and $b$ (blue–yellow), which describe the chromatic distribution and saturation characteristics in a perceptually uniform manner.

The gamut visualizations in Figure 4 reveal clear improvements in color range, chromatic distribution, and perceptual uniformity after applying the HSI-based correction. In the night scene, the

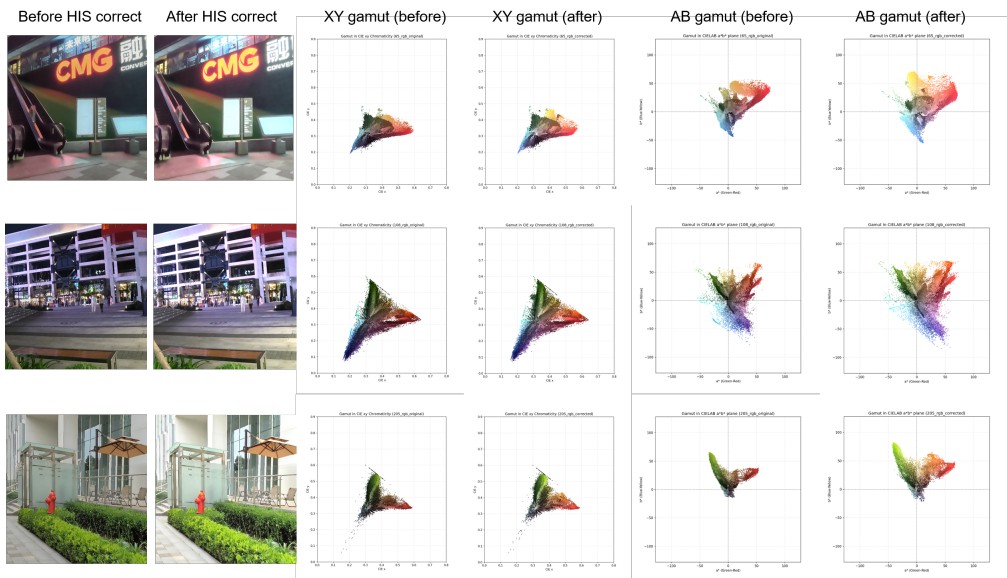

Figure 4: Visual comparison on XY and AB CIE space on diverse real-world scenes.

neon red and blue lights regain distinct chromatic signatures, reducing hue mixing visible before the correction. In the outdoor scenes, vegetation points shift from a yellow-green cluster to a more natural green distribution with higher negative $a$ values. The xy gamut plots show that the correction expands chromaticity coverage from a desaturated central cluster to a fuller distribution toward the RGB primaries, indicating restored saturation and more accurate color rendering. The Lab plots reveal that the correction transforms a yellow-biased, compressed color distribution into a well-balanced, quadrant-spanning gamut with richer reds, greens, and blues, demonstrating improved hue diversity.

## 6 CONCLUSION

This work addresses the critical quality bottleneck in wearable ego-centric imaging by tackling core challenges: limited paired data, distortion mismatches, and poor color fidelity. Our solution offers three main contributions: we introduce HHW-Ego, the first multi-source dataset integrating HSI, HDR, and wearable imagery; we implement a customized degradation pipeline that generates realistic low-quality samples, enabling supervised learning; and we propose a two-stage framework that uses HHW-Ego for detail recovery and HSI for crucial color temperature refinement. Extensive experiments validate that our method consistently improves wearable image quality, narrowing the gap with high-end cameras. Future work will focus on expanding HHW-Ego to extreme scenarios, optimizing the framework for real-time wearable inference, and integrating it with downstream embodied intelligence tasks.

## ETHICS STATEMENT

The authors have read and adhere to the ICLR Code of Ethics. This work does not involve human subjects, identifiable private data, or harmful applications. All datasets used are publicly available and were used in accordance with their original licenses and intended purposes. No external sponsorship or conflict of interest influenced the design or conclusions of this work.

## REPRODUCIBILITY STATEMENT

All code and source files are provided in the supplementary material and will be publicly released. Additional experiments can be found in Appendix A.

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

# A ADDITIONAL EXPERIMENTS

## A.1 ABLATION EXPERIMENTS

Our scheme is compatible with different ISP processing pipelines. EmdoorVR uses images processed by a standard ISP, while RayNeo features an ISP algorithm specifically tuned for its AR1 chip. Thus, when comparing images output by these two glasses, we can observe that RayNeo's images are of slightly higher quality than EmdoorVR's. However, with the integration of our scheme, even better image quality can be achieved.

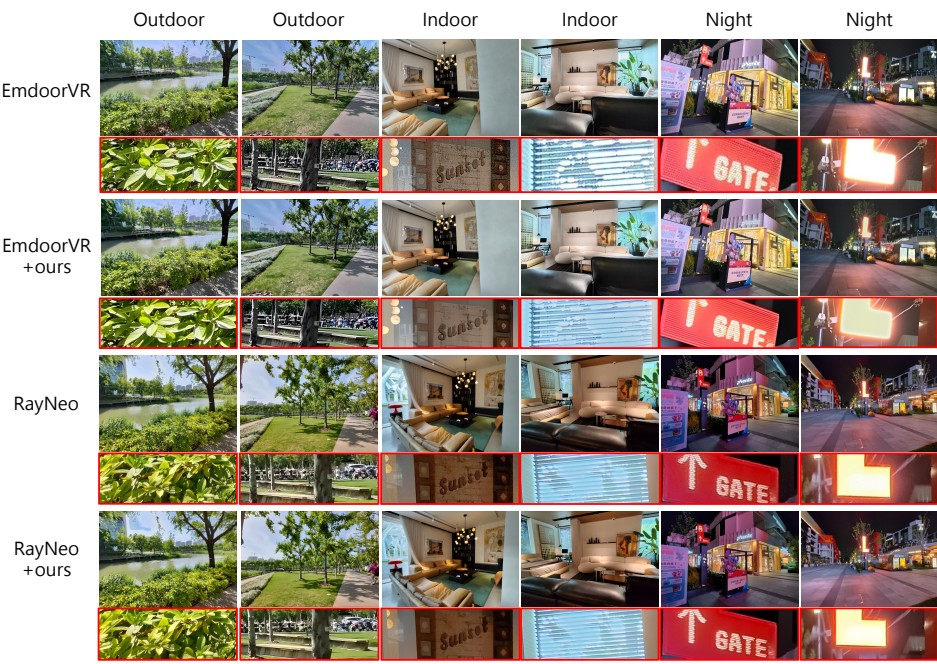

Figure 5: Comparison of our method across different scenarios (Outdoor, Indoor, Night) and data sources (EmdoorVR, RayNeo)

## B  THE USE OF LARGE LANGUAGE MODELS (LLMs)

We disclose that we used Qwen3-Max-Preview(Yang et al., 2025a) to assist in polishing the language and improving the clarity of this paper. The model was used for grammar correction, sentence restructuring, and enhancing overall readability. All technical content, experimental design, results, and conclusions were authored and verified solely by the human authors. The LLM did not contribute to the generation of ideas, methods, or data analysis.

