# OpenReview forum: "HHW-Ego: DSLR-Quality Enhancement for Multi-Source Wearable Ego Imaging"
_ICLR.cc/2026/Conference — ICLR 2026 Conference Withdrawn Submission_

### Official Review · Reviewer_Fehf · 2025-10-21

**Soundness:** 3
**Presentation:** 2
**Contribution:** 3
**Rating:** 4
**Confidence:** 3

**Summary:**

This paper addresses wearable Ego imaging. The authors design a customized degradation pipeline to generate paired samples that match wearable camera images, with the goal of raising the upper bound of enhancement performance. In addition, the authors construct a two‑stage enhancement framework and introduce HHW‑Ego, a multi‑source paired dataset that fuses hyperspectral data, high dynamic range, and wearable first‑person imagery.

**Strengths:**

1. The paper contributes HHW‑Ego, a relatively rare multi‑source dataset. The data collection pipeline is described in detail and appears credible.

2. The use of hyperspectral data for further color correction is supported by relatively in‑depth theoretical analysis.

3. The presentation is nice.

**Weaknesses:**

1. The approach used in the first stage is similar to existing super‑resolution models (e.g., Real-ESRGAN). The authors claim the novelty lies in adding LAB correction in the data pipeline, but the reviewer finds this insufficiently novel. The authors should clarify the uniqueness and suitability of this modification specifically for wearable imaging.

2. In the last row of Figure 3, the results produced by the method appear to exhibit color shifts (see the lighting region).

3. In the main Table 2, the method does not seem to improve SSIM and PIQE. On the contrary, both metrics degrade after applying the method. The authors are encouraged to analyze the reasons for this.

4. The additional prior from HSI data is presented as the main innovation of the method. However, in the second row of Table 3 this leads to performance degradation; the reviewer considers the corresponding gain to be marginal.

An execllent response will rise the score.

**Questions:**

Please refer to weaknesses.

---

### Official Review · Reviewer_PtW8 · 2025-10-30

**Soundness:** 3
**Presentation:** 1
**Contribution:** 2
**Rating:** 4
**Confidence:** 4

**Summary:**

A custom degradation pipeline generates matching paired samples, raising enhancement limits.
A two-stage enhancement framework is proposed.
HHW-Ego is proposed, a multi-source paired dataset integrating Hyperspectral, HDR, and Wearable data.
Experiments show the framework effectively enhances images.

**Strengths:**

1. Customized Degradation Pipeline for Paired Data Creation.

2. The paper introduces a novel two-stage enhancement framework.

3. Introduction of HHW-Ego: A Comprehensive Multi-Source Dataset.

4. Writing is easy to follow.

**Weaknesses:**

1. It is not clearly stated whether the baseline models in Table 2 were trained on wearable camera datasets or used as official pre-trained models.

2. In Table 2, the "Ours" results should be placed closer to the corresponding baselines for easier comparison. Better visual cues, such as bolding the best results, would also improve readability (Same in Table 3).

3. The paper mentions terms like color fidelity and spectral consistency without clear definitions or explanations. These concepts should be explained, especially for readers unfamiliar with them.

4. The paper claims improvements in color fidelity and spectral consistency but does not provide visual examples to support these claims. Comparative images would help validate the effectiveness of the method. The conclusion in Section 5.5 (e.g., Lines 433–436) is not strongly supported. It claims color fidelity improvements from hyperspectral integration, but only shows numerical results in Table 3 without a clear visual presentation.

5. The introduction does not clearly explain why integrating hyperspectral and HDR data helps improve color fidelity, spectral consistency, and detail. The motivation feels disjointed and unclear.

**Questions:**

1. In Figure 3, what baseline method is used for comparison with "Ours"? Please clarify.

2. When computing SSIM, what does "the original input" refer to? Is it the low-quality input?

3. Can you create some paired test data using the same degradation method as the training data to make the evaluation more reliable?

4. The metrics used are insufficient. Could you add more representative no-reference metrics, such as MUSIQ?

5. There is a grammatical issue in Lines 447–449. Please revise the sentence for clarity.

---

### Official Review · Reviewer_AsvG · 2025-11-01

**Soundness:** 2
**Presentation:** 2
**Contribution:** 3
**Rating:** 4
**Confidence:** 3

**Summary:**

This paper addresses wearable ego-centric image enhancement by proposing a two-stage framework that leverages a customized degradation pipeline for data generation and hyperspectral-guided color refinement, supported by a novel multi-source paired dataset (HHW-Ego) of hyperspectral, HDR, and wearable data.

**Strengths:**

1. This work tackles a problem of significant and growing practical importance: the quality gap between wearable ego-centric cameras and consumer-grade DSLR/cellphone cameras. As wearable devices become ubiquitous for first-person documentation and embodied AI, this research directly addresses a critical bottleneck in visual quality, with clear applications in augmented reality, assistive technology, and lifelogging.

2. A key strength is the creation of the HHW-Ego dataset, which fills a critical gap in the community. By providing meticulously paired hyperspectral, HDR, and wearable data across diverse scenes, it offers an invaluable resource not only for supervised image enhancement but also for fundamental research in color science and illumination modeling under real-world conditions.

3. The proposed two-stage framework presents a pragmatic and well-structured solution. The combination of a data degradation pipeline with a spectral-based color refinement module constitutes a solid engineering approach for this task.

**Weaknesses:**

**1. Lack of Critical Implementation Details**

The manuscript omits several key technical details, which hinders reproducibility and a full understanding of the proposed method.

**Unexplained Module**: The exposure correction module, which is first mentioned and evaluated in the ablation study (Section 5.5), is not described in the methodology section. Its role and implementation remain unclear to the reader.

**Unspecified Image Processing Pipeline**: The description of the data processing pipeline is ambiguous. It is critical to clarify whether the proposed data preprocessing and enhancement framework operates directly on RAW data or on post-processed sRGB images generated by a separate ISP.

**Incomplete Figure Annotation**: Figure 1 is cited as containing "red solid lines" and "black dashed lines," which are not present in the provided figure, leading to confusion.

**Undefined Hyperparameter**: The introduction of the "strength" parameter in the CCT map calculation (Line 249) is not sufficiently motivated. Its impact on the final results and the rationale for its specific value are not discussed.

**2. Insufficient Evidence for Key Claims**

The experimental evidence does not yet solidly support some of the paper's central claims regarding its unique contributions.

**Necessity of Color Alignment**: The authors claim that a customized degradation pipeline is a core contribution, with the unique step being color alignment in Lab space. However, the performance gain from this step appears marginal in Table 4. The authors should provide stronger ablation results or analysis to conclusively demonstrate its necessity and unique benefit over other common color jittering techniques.

**Benefit of Hyperspectral Guidance**: The claim that the hyperspectral-guided stage enhances color fidelity is not fully validated.

The evaluation relies solely on general image quality metrics (e.g., PSNR, SSIM), which are not sensitive enough to isolate improvements in color accuracy.

The visual comparisons provided do not clearly demonstrate superior color reproduction compared to the baselines.

To substantiate this claim, the ablation study (Table 3) should be supplemented with color-specific metrics and targeted visual examples that unequivocally show the color improvement brought by the hyperspectral data.

**Questions:**

1. What's the implement of exposure correction module ?
2. Please clarify whether the proposed data preprocessing and enhancement framework operates directly on RAW data or on post-processed sRGB images generated by a separate ISP.
3. What's the value and effect of the "strength" parameter in the CCT map calculation (Line 249) ?
4. Any other stronger justification to validate the necessity of color alignment in degradation pipeline?
5. Any further evidence to validate the Benefit of Hyperspectral Guidance?

---

### Note · Authors · 2026-02-26

I have read and agree with the venue's withdrawal policy on behalf of myself and my co-authors.

---

### Meta-Review · Area_Chair_YpTb · 2025-12-13

**Summary:**

The decision to reject this paper is driven by collective critical concerns from three reviewers, which comprehensively undermine the paper’s credibility, validity, and contribution to wearable ego-centric image enhancement research:
1. The paper omits critical implementation information essential for reproducibility and understanding, including the design and role of the exposure correction module (only mentioned in ablation but not methodology), ambiguity about whether the framework operates on RAW data or post-processed sRGB images, undefined "strength" hyperparameter in CCT map calculation (no motivation or impact analysis), and confusing figure annotations (e.g., non-existent "red solid lines" in Figure 1).

2. The paper’s central assertions—such as the benefit of hyperspectral guidance for color fidelity and the necessity of Lab-space color alignment in the degradation pipeline—are not adequately validated. Evaluations rely solely on general quality metrics (e.g., PSNR, SSIM) without color-specific metrics or targeted visual examples to confirm color improvement. Moreover, the claimed advantage of hyperspectral integration shows marginal gains or even performance degradation in ablation results, while the marginal gain from Lab color alignment fails to justify its necessity.

3. Key experimental results are unconvincing or unexplained—for instance, the proposed method degrades SSIM and PIQE metrics in main experiments without analysis, and produces color shifts in visual results. Tables lack clear visual cues for comparison (e.g., unbolded best results) and improper layout, while critical terms (e.g., color fidelity, spectral consistency) are undefined. Additionally, the baseline training settings are unclear, and the evaluation lacks reliable paired test data and representative no-reference metrics (e.g., MUSIQ).

4. The first-stage approach closely resembles existing super-resolution models (e.g., Real-ESRGAN), and the claimed novelty of adding Lab correction in the data pipeline is insufficiently justified, with no clarification of its uniqueness or suitability for wearable imaging scenarios.

**Reviewer Concerns:**

Since the author did not provide a rebuttal to resolve these issues, all the concerns remain.

**Reviewer Scores:**

Since the author did not provide a rebuttal to resolve these issues, all the concerns remain.

---

### Decision · Program_Chairs · 2026-01-26

Reject